# Evaluation of low-dose aspirin in the prevention of recurrent spontaneous preterm labour (the APRIL study): A multicentre, randomised, double-blinded, placebo-controlled trial

Anadeijda J. E. M. C. Landman [1]*, Marjon A. de Boer[1], Laura Visser[1], Tobias A. J. Nijman[2], Marieke A. C. Hemels[3], Christiana N. Naaktgeboren [4], Marijke C. van der Weide [4], Ben W. Mol [5,6], Judith O. E. H. van Laar[7], Dimitri N. M. Papatsonis[8], Mireille N. Bekker [9], Joris van Drongelen [10], Mariëlle G. van Pampus[11], Marieke Sueters[12], David P. van der Ham[13], J. Marko Sikkema[14], Joost J. Zwart[15], Anjoke J. M. Huisjes[16], Marloes E. van Huizen[17], Gunilla Kleiverda[18], Janine Boon[19], Maureen T. M. Franssen[20], Wietske Hermes[2], Harry Visser [21], Christianne J. M. de Groot[1], Martijn A. Oudijk[4]

1 Department of Obstetrics and Gynaecology, Amsterdam UMC, Vrije Universiteit Amsterdam, Amsterdam Reproduction & Development Research Institute, Amsterdam, the Netherlands, 2 Department of Obstetrics and Gynaecology, Haaglanden Medical Centre, Den Haag, the Netherlands, 3 Department of Neonatal Intensive Care, Isala, Zwolle, the Netherlands, 4 Department of Obstetrics and Gynaecology, Amsterdam UMC, University of Amsterdam, Amsterdam Reproduction & Development Research Institute, Amsterdam, the Netherlands, 5 Department of Obstetrics and Gynaecology, School of Clinical Sciences at Monash Health, Monash University, Melbourne, Victoria, Australia, 6 Aberdeen Centre for Women's Health Research, University of Aberdeen Aberdeen, United Kingdom, 7 Department of Obstetrics and Gynaecology, Máxima Medical Centre, Veldhoven, the Netherlands, 8 Department of Obstetrics and Gynaecology, Amphia Hospital, Breda, the Netherlands, 9 Department of Obstetrics and Gynaecology, University Medical Centre Utrecht, Utrecht, the Netherlands, 10 Department of Obstetrics and Gynaecology, Radboud University Medical Center, Nijmegen, the Netherlands, 11 Department of Obstetrics and Gynaecology, OLVG, Amsterdam, the Netherlands, 12 Department of Obstetrics and Gynaecology, Leiden University Medical Centre, Leiden, the Netherlands, 13 Department of Obstetrics and Gynaecology, Martini Hospital, Groningen, the Netherlands, 14 Department of Obstetrics and Gynaecology, Hospital Group Twente Almelo, Almelo, the Netherlands, 15 Department of Obstetrics and Gynaecology, Deventer Hospital, Deventer, the Netherlands, 16 Department of Obstetrics and Gynaecology, Gelre Hospitals Apeldoorn, Apeldoorn, the Netherlands, 17 Department of Obstetrics and Gynaecology, Haga Hospital, Den Haag, the Netherlands, 18 Department of Obstetrics and Gynaecology, Flevo Hospital Almere, Almere, the Netherlands, 19 Department of Obstetrics and Gynaecology, Diakonessenhuis, Utrecht, the Netherlands, 20 Department of Obstetrics and Gynaecology, University Medical Centre Groningen, Groningen, the Netherlands, 21 Department of Obstetrics and Gynaecology, Tergooi Hospitals, Hilversum, the Netherlands

* a.landman@amsterdamumc.nl

## Abstract

### Background

Preterm birth is the leading cause of neonatal morbidity and mortality. The recurrence rate of spontaneous preterm birth is high, and additional preventive measures are required. Our objective was to assess the effectiveness of low-dose aspirin compared to placebo in the prevention of preterm birth in women with a previous spontaneous preterm birth.

**Data Availability Statement:** Data are owned by Amsterdam University Medical Centers, location Academic Medical Center (AMC), that supports data sharing. For enquiries on data sharing, a methodologist from the trial centre within the Dutch Consortium for Healthcare Evaluation and Research in Obstetrics and Gynaecology (NVOG Consortium) may be contacted via trialbureau@zorgevaluatienederland.nl. The de-identified data underlying this study will be made available for research purposes. The requestors are required to sign a data sharing agreement to ensure appropriate use of the data, and to ensure data confidentiality and security.

**Funding:** MdB, LV, TN, BM, MO and CdG received funding from ZonMw, The Dutch Organisation for Health Research and Development (grant number 836041006). The funders had no role in study design, data collection and analysis, decision to publish, or preparation of the manuscript.

**Competing interests:** I have read the journal's policy and the authors of this manuscript have the following competing interests: BM reported an Investigator grant from the National Health and Medical Research Council (NHMRC; grant no. GNT1176437); receipt of research funding from Guerbet; and is a former advisory board member at ObsEva. All other authors do not report any relevant financial activities outside the submitted work.

**Abbreviations:** AMC, Academic Medical Center; CI, confidence interval; CONSORT, Consolidated Standards of Reporting Trials; HELLP, hemolysis, elevated liver enzymes, and low platelets; NICU, neonatal intensive care unit; RR, relative risk.

## Methods and findings

We performed a parallel multicentre, randomised, double-blinded, placebo-controlled trial (the APRIL study). The study was performed in 8 tertiary and 26 secondary care hospitals in the Netherlands. We included women with a singleton pregnancy and a history of spontaneous preterm birth of a singleton between 22 and 37 weeks. Participants were randomly assigned to aspirin 80 mg daily or placebo initiated between 8 and 16 weeks of gestation and continued until 36 weeks or delivery. Randomisation was computer generated, with allocation concealment by using sequentially numbered medication containers. Participants, their healthcare providers, and researchers were blinded for treatment allocation. The primary outcome was preterm birth <37 weeks of gestation. Secondary outcomes included a composite of poor neonatal outcome (bronchopulmonary dysplasia, periventricular leukomalacia > grade 1, intraventricular hemorrhage > grade 2, necrotising enterocolitis > stage 1, retinopathy of prematurity, culture proven sepsis, or perinatal death). Analyses were performed by intention to treat.

From May 31, 2016 to June 13, 2019, 406 women were randomised to aspirin ($n = 204$) or placebo ($n = 202$). A total of 387 women (81.1% of white ethnic origin, mean age 32.5 ± SD 3.8) were included in the final analysis: 194 women were allocated to aspirin and 193 to placebo. Preterm birth <37 weeks occurred in 41 (21.2%) women in the aspirin group and 49 (25.4%) in the placebo group (relative risk (RR) 0.83, 95% confidence interval (CI) 0.58 to 1.20, $p = 0.32$). In women with $\geq$80% medication adherence, preterm birth occurred in 24 (19.2%) versus 30 (24.8%) women (RR 0.77, 95% CI 0.48 to 1.25, $p = 0.29$). The rate of the composite of poor neonatal outcome was 4.6% ($n = 9$) versus 2.6% ($n = 5$) (RR 1.79, 95% CI 0.61 to 5.25, $p = 0.29$). Among all randomised women, serious adverse events occurred in 11 out of 204 (5.4%) women allocated to aspirin and 11 out of 202 (5.4%) women allocated to placebo. None of these serious adverse events was considered to be associated with treatment allocation. The main study limitation is the underpowered sample size due to the lower than expected preterm birth rates.

## Conclusions

In this study, we observed that low-dose aspirin did not significantly reduce the preterm birth rate in women with a previous spontaneous preterm birth. However, a modest reduction of preterm birth with aspirin cannot be ruled out. Further research is required to determine a possible beneficial effect of low-dose aspirin for women with a previous spontaneous preterm birth.

## Trial registration

Dutch Trial Register (NL5553, NTR5675) https://www.trialregister.nl/trial/5553.

---

## Author summary

### Why was this study done?

- Complications of preterm birth are the leading cause of neonatal morbidity and mortality worldwide.

- As spontaneous preterm birth has a multifactorial etiology, different preventive measures, in addition to, e.g., progesterone, may be needed depending on the underlying cause in individual cases.

- Due to the partially similar pathophysiology of uteroplacental ischemia between spontaneous preterm birth and preeclampsia, it has been suggested that low-dose aspirin may also prevent spontaneous preterm birth.

## What did the researchers do and find?

- We performed a multicentre, randomised, double-blinded, placebo-controlled trial including 387 women with a singleton pregnancy and a previous spontaneous preterm birth of a singleton.

- Women were allocated to aspirin 80 mg or placebo starting between 8 and 16 weeks of gestation and continued until 36 weeks of gestation.

- The preterm birth rate was slightly lower in the aspirin group (21.2%) as compared to the placebo group (25.4%), especially among women with good adherence to medication (19.2% in the aspirin group versus 24.8% in the placebo group). These differences were not statistically significant.

## What do these findings mean?

- The study sample was too small to draw firm conclusions on the effectivity of low-dose aspirin for the prevention of spontaneous preterm birth, and, therefore, we do not recommend implementation of low-dose aspirin for this indication at this time.

- Further research is necessary to evaluate the possible beneficial effect of low-dose aspirin on the prevention of spontaneous preterm birth.

## Introduction

Worldwide approximately 15 million preterm births occur each year, which accounts for 9% to 12% of all live births [1]. Complications of preterm birth are the leading cause of perinatal mortality and mortality among children younger than 5 [2]. Surviving neonates are at increased risk for several short-term and long-term morbidities such as neurodevelopmental and cognitive impairment [3–5].

Approximately 65% of preterm births has a spontaneous onset following contractions or the rupture of membranes [6]. The strongest risk factor for spontaneous preterm birth is a previous spontaneous preterm birth, with a recurrence rate of up to 30% to 35% in singleton pregnancies [7]. Even when preventive measures such as progesterone are applied, many women will deliver preterm again in subsequent pregnancies [8]. Additional preventive strategies are, therefore, urgently needed.

Low-dose aspirin has been proven effective for the prevention of preeclampsia and has also shown to reduce the overall rate of preterm birth [9,10]. In earlier studies, the reduction of preterm birth following the use of low-dose aspirin was attributed to a reduction of medically indicated preterm birth related to preeclampsia. Due to the partially similar pathophysiology

of uteroplacental ischemia between spontaneous preterm birth and preeclampsia, it has been suggested that low-dose aspirin may also prevent spontaneous preterm birth. Secondary analyses of randomised controlled trials and a large individual patient data meta-analysis of women at risk for preeclampsia have indeed shown evidence that low-dose aspirin may reduce spontaneous preterm birth [11–13]. To the best of our knowledge, no trials have evaluated aspirin in a high-risk population of women with a history of spontaneous preterm birth.

The aim of the present study is to assess the effectiveness of low-dose aspirin in comparison with placebo in preventing preterm birth when initiated in early pregnancy in a high-risk population of women with a previous spontaneous preterm birth.

## Methods

### Study design

The low-dose aspirin in the prevention of recurrent spontaneous preterm Labour (APRIL) study was a multicentre, randomised, double-blinded, placebo-controlled trial comparing aspirin 80 mg daily to placebo for the prevention of recurrent preterm birth in a high-risk population of women with a previous spontaneous preterm birth. We conducted the trial in 8 tertiary and 26 secondary care hospitals in the Netherlands within the Dutch Consortium for Healthcare Evaluation and Research in Obstetrics and Gynaecology (NVOG Consortium). Ethics approval was obtained from the Medical Research Ethics Committee from the Amsterdam Medical Center (no. 2015_332) and by the boards of all participating centres. The trial was registered in the Dutch Trial Register (NL5553, NTR5675). The study protocol has been published [14]. Minor changes to the study protocol were made before completion of the trial (S1 Appendix). This study is reported as per the Consolidated Standards of Reporting Trials (CONSORT) checklist (S2 Appendix) [15].

### Participants

Women ≥18 years were eligible for participation when they had a singleton pregnancy between 8 and 16 weeks of gestation and had a previous spontaneous preterm birth of a singleton between 22 and 37 weeks of gestation. Spontaneous preterm birth was defined as preterm birth following spontaneous contractions with intact membranes or preterm birth after spontaneously ruptured membranes. Exclusion criteria were other indications for aspirin administration (determined at the discretion of the healthcare providers in the participating centres), thrombocytopenia or thrombocytopathy, and major fetal malformations in the current pregnancy or a previous pregnancy ending in spontaneous preterm birth.

### Randomisation and masking

Eligible women were recruited by healthcare providers or research nurses/midwives and were given written information about the trial. Written informed consent was obtained from all participants. Women were randomised in a 1:1 ratio using random permuted blocks of sizes 2 and 4 without stratification or minimisation. We used ALEA, a web-based interface that displays the allocation from a computer-generated randomisation sequence. Participants, their healthcare providers, and researchers were blinded for treatment allocation. Allocation deblinding of the research team was performed after the completion of data collection.

### Intervention

Women were allocated to aspirin 80 mg or matched placebo. Tablets were manufactured, packaged, and labelled by Ace Pharmaceuticals in the Netherlands. Placebo and aspirin tablets

were identical with respect to size, appearance, and physical properties. Medication containers were of identical appearance, sealed, and numbered sequentially according to the allocation sequence. Study medication was stored at the trial pharmacy of Amsterdam University Medical Centers, location Academic Medical Center (AMC). After randomisation, women were prescribed the study medication, which was then distributed to the participant's preferred address. Treatment was initiated between 8 and 16 weeks of gestation and was continued until 36 weeks or delivery, if delivery occurred earlier. Women were instructed to take 1 tablet a day, preferably in the evening. Other preventive interventions for the prevention of preterm birth, such as progesterone, cerclage, or pessary, could be used alongside the study.

## Outcomes

The primary outcome was preterm birth ≥16 and <37 weeks of gestation. Furthermore, we assessed preterm birth rates ≤34 and ≤28 weeks of gestation. Preterm birth was subdivided into spontaneous and indicated preterm birth. Secondary perinatal outcomes were gestational age at birth, preterm prelabour rupture of membranes (with delivery <37 weeks of gestation), midtrimester fetal loss ($16^{+0}$ to $21^{+6}$ weeks of gestation), mode of birth, postpartum hemorrhage, birth weight, and small for gestational age (<10th centile) [16]. Secondary neonatal outcomes included a composite of poor neonatal outcome containing bronchopulmonary dysplasia, periventricular leucomalacia > grade 1, intraventricular hemorrhage > grade 2, necrotising enterocolitis > stage 1, retinopathy of prematurity, culture-proven sepsis, and perinatal death. The components of the composite of poor neonatal outcome were also analysed separately. Other neonatal outcomes were the number and days of hospital admissions or admissions to the neonatal intensive care unit (NICU). Maternal secondary outcomes included mortality, gestational diabetes, pregnancy-induced hypertension, preeclampsia/hemolysis, elevated liver enzymes, and low platelets (HELLP) syndrome, eclampsia, pulmonary edema, thromboembolic disease, placental abruption, vaginal bleeding, other bleeding, gastrointestinal complaints, hospital admissions (total number of days and number of admissions for vaginal bleeding or threatened preterm labour), course of steroids for fetal lung maturity, tocolytic therapy, and interventions during pregnancy (progesterone, cerclage, pessary, bacterial vaginosis treated with antibiotics, and urinary or genital tract infections treated with antibiotics). Definitions of the secondary outcomes are provided in S3 Appendix. We used the core outcomes for research concerning interventions to prevent preterm birth [17].

Serious adverse events were defined as congenital anomalies or events that resulted in maternal, fetal, or neonatal death; were life threatening; required hospitalisation (for complications that were not inherent to pregnancy); resulted in persistent or significant disability or incapacity; or any other serious or unexpected adverse event.

## Data collection and medication adherence

Gestational age was determined by first trimester ultrasound according to the Dutch national guidelines [18]. Participants and their infants were followed up until 3 months from the expected date of delivery. Maternal characteristics; medical and obstetrical histories; and pregnancy, birth, and neonatal outcomes were recorded in an electronic case report form. Participants received diaries to track their medication use and to report possible side effects from the medication. Symptoms were scored from 0 (no imposition) to 5 (severe imposition). In addition, participants were requested to return leftover study medication to the hospital for pill counts. We calculated medication adherence by dividing the number of used tablets by the expected number of doses per participant. Good adherence was defined as

tablet intake of ≥80%. A detailed description of the medication adherence calculation is provided in S4 Appendix.

## Statistical analysis

In the sample size calculation, we estimated that aspirin could potentially reduce the recurrent preterm birth rate from 36% to 23%, which is a reduction of 35%. We calculated that enrollment of 384 women was needed (2-sided $\alpha = 0.05$, $\beta = 80\%$). Taking into account 5% loss to follow-up, the target enrollment number was 406 women: 203 in each treatment arm.

A statistical analysis plan was developed and finalised before deblinding of treatment allocation (S5 Appendix). No changes were made in the planned analyses. We performed analyses by intention to treat. For dichotomous outcomes, treatment effects within the aspirin group were quantified as relative risks (RRs) as calculated by generalised linear regression analysis using a log link. The corresponding 95% confidence intervals (CIs) and $p$-values were also presented. When there were ≤5 events for a variable, Fisher exact test was used to calculate the $p$-value instead of the generalised linear regression model. For outcomes with multiple categories, we performed a chi-squared test. For visually normally distributed continuous outcomes, we calculated mean differences with the corresponding 95% CIs and $p$-values by using the independent samples $t$ test. For highly skewed continuous outcomes, medians and interquartile ranges were calculated together with the 95% CIs and differences in medians. We reported the $p$-value from the Mann–Whitney U test. No covariates were adjusted for in the analyses. For the primary outcome, Bayes factor analysis was conducted using an RR of 0.64 as prior (based on the assumptions given in the sample size calculation) and a normal distribution for the hypothesis under test.

Prespecified subgroup analyses were conducted by adding an interaction term and testing for its statistical significance. The prespecified subgroup analyses were gestational age at initiation of treatment ($8^{+0}$ to $11^{+6}$ weeks versus $12^{+0}$ to $16^{+0}$ weeks); progesterone prophylaxis versus no progesterone; cervical length (<25 mm versus ≥25 mm); gestational of the earliest previous spontaneous preterm birth (<30+0 weeks versus $30^{+0}$ to $33^{+6}$ weeks and $34^{+0}$ to $36^{+6}$ weeks); and the onset of the earliest previous spontaneous preterm birth (spontaneous contractions versus preterm prelabour rupture of membranes). We also performed a prespecified sensitivity analysis of the primary outcome, including women with ≥80% adherence with medication. The cumulative incidence of preterm birth according to treatment group was illustrated in a Kaplan–Meier plot. Statistical analyses were performed in SPSS version 26.0 and SAS version 9.4. We used a 2-sided $p$-value of 0.05. No corrections were made for multiple comparisons. An external Data Safety and Monitoring Committee monitored safety after every 100th inclusion. No interim analyses for efficacy were performed. The planned economic analysis as described in the protocol paper will be reported elsewhere [14].

## Patient and public involvement

The trial was designed in cooperation with the Dutch association for parents with a preterm born child (Care4Neo), and they fully endorsed the research proposal upon submission to ZonMw, the funding organisation. For outcome measures, we used the core outcome set for evaluation of interventions to prevent preterm birth, which was developed in collaboration with patient representatives [17]. We plan to disseminate the results to study participants and patient organisations such as Care4Neo.

## Results

A total of 406 women were randomised from May 31, 2016 to June 13, 2019. Follow-up was completed on March 27, 2020 (Fig 1). The exact number of screened women and women who declined participation is unknown, as participating centres were not allowed to collect and share these data according to the Dutch privacy regulations. After randomisation, we excluded 4 women with terminations of pregnancy due to genetic abnormalities (2 in the aspirin group versus 2 in the placebo group), 5 with major congenital anomalies (2 versus 3), and 10 women were ineligible (6 versus 4). Women were ineligible if they were included based on a midtrimester fetal loss <22 weeks of gestation (*n* = 6), if the previous preterm birth concerned multiple

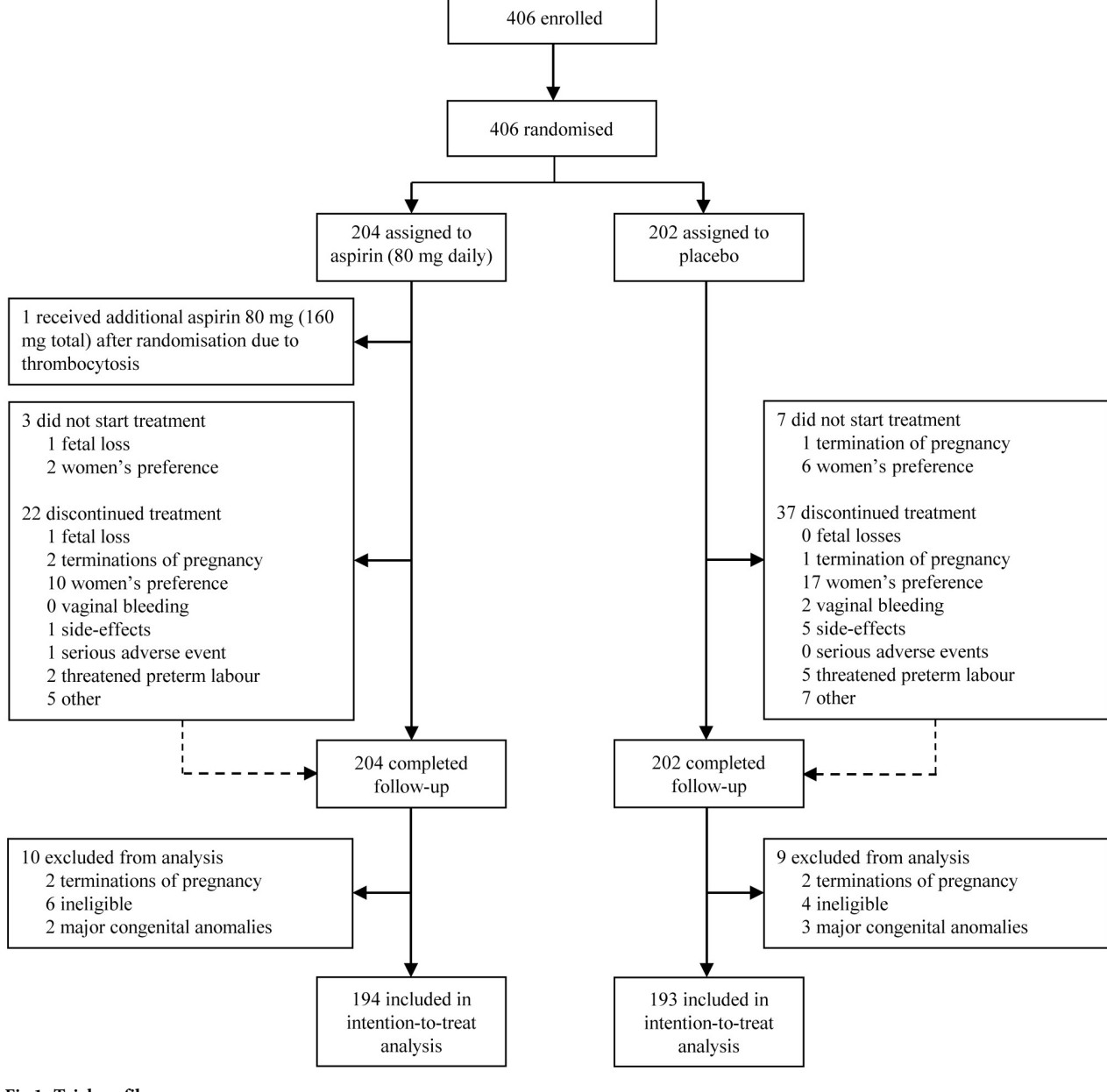

**Fig 1. Trial profile.**

gestations ($n = 2$) or a child with congenital anomalies ($n = 1$), or if women had another indication for aspirin at baseline ($n = 1$). A total of 387 women were included in the final analysis: 194 were allocated to aspirin and 193 to placebo. The inclusions per participating centre are illustrated in S1 Table.

The median gestational age at randomisation was $13^{+5}$ weeks (IQR $12^{+1}$ to $15^{+0}$) in the aspirin group and $13^{+6}$ weeks (IQR $11^{+6}$ to $15^{+1}$) in the placebo group. Baseline characteristics between treatment groups were comparable regarding maternal characteristics and maternal medical history (Table 1). The rate of women with 1 or more previous midtrimester fetal losses was higher in the aspirin group ($n = 14$, 7.2%) than in the placebo group ($n = 2$, 1.0%). These women had multiple preterm births, as a history of a spontaneous preterm birth $\geq 22$ weeks of gestation was an inclusion criterion. Additionally, women in the aspirin group more often had a history of cervical surgery, a history of uterine surgery, and a positive family history of preterm birth.

Median gestational age at initiation of study medication was $14^{+5}$ (IQR $13^{+3}$ to $15^{+6}$) for women allocated to aspirin and $15^{+1}$ (IQR $12^{+6}$ to $16^{+0}$) for women allocated to placebo (S2 Table). Medication adherence was $\geq 80\%$ in 63.3% of women, $<80\%$ in 11.1%, and unknown in 25.6%. There were no differences in medication adherence between study groups.

Preterm birth $<37$ weeks of gestation occurred in 41 (21.1%) women allocated to aspirin and in 49 (25.4%) women allocated to placebo (RR 0.83, 95% CI 0.58 to 1.20, $p = 0.32$) (Table 2). The Bayes factor was 0.97, indicating no strong evidence of difference between arms. In the prespecified sensitivity analysis including women with good adherence ($\geq 80\%$) to medication, preterm birth $<37$ weeks of gestation occurred less frequent in the aspirin group (24 women, 19.2%) as compared to the intention-to-treat population, while the preterm birth rate in the placebo group remained approximately the same (30 women, 24.8%). The difference between treatment groups was not statistically significant (RR 0.77, 95% CI 0.48 to 1.25, $p = 0.29$). There was no difference in total, spontaneous, or indicated preterm birth $<37$ weeks of gestation between women allocated to aspirin or placebo. The cumulative percentages of women with a preterm birth $<37$ weeks of gestation did not differ (log-rank $p = 0.36$) between treatment groups (Fig 2). Preterm prelabour rupture of membranes occurred in 9 (4.6%) women receiving aspirin and in 18 (9.3%) receiving placebo (RR 0.50, 95% CI 0.23 to 1.08, $p = 0.077$). There were no differences between treatment groups for gestational age at birth, preterm birth $\leq 34$ and $\leq 28$ weeks of gestation, mode of delivery, postpartum hemorrhage, and small for gestational age neonates.

The composite poor neonatal outcome occurred in 9 (4.6%) neonates in the aspirin group and 5 (2.6%) in the placebo group (RR 1.79, 95% CI 0.61 to 5.25, $p = 0.29$) (Table 3). There were no significant differences between the separate components of the composite outcome. A total of 6 deaths occurred in the aspirin group: 2 fetal losses diagnosed around 16 weeks of gestation (of which one had not started medication yet), 2 midtrimester fetal losses ($18^{+6}$ and $21^{+4}$ weeks), and 2 extreme preterm births ($24^{+2}$ and $25^{+2}$ weeks). There were 2 deaths in the placebo group, which were both midtrimester fetal losses ($22^{+5}$ and $23^{+4}$ weeks). There were no differences in length of NICU or hospital admissions until 3 months corrected neonatal age. Additional neonatal outcomes can be found in S3 Table.

There were no differences in the incidence of maternal hypertensive disorders, other morbidities, or hospital admissions during pregnancy between treatment groups (Table 4). More women in the aspirin group received a cerclage during pregnancy ($n = 25$, 12.9%) compared to the placebo group ($n = 11$, 5.7%) (RR 2.26, 95% CI 1.15 to 4.47, $p = 0.019$). The use of other preventative measures for preterm birth and maternal self-reported symptoms were similar between treatment groups. Additional data on maternal self-reported symptoms can be found in S1 Fig.

**Table 1. Baseline characteristics according to treatment group.**

| | Aspirin (*n* = 194) | Placebo (*n* = 193) |
|---|---|---|
| Age (years)–mean (SD) | 32.8 (±3.9) | 32.3 (±3.6) |
| Body mass index (kg/m$^2$)–median (IQR) | 23.8 (21.6 to 26.7) | 23.7 (21.5 to 26.9) |
| Ethnic origin | | |
| White | 154/182 (84.6%) | 160/185 (86.5%) |
| Other origins | 28/182 (15.4%) | 25/185 (13.5%) |
| Education | | |
| Low[a] | 4/108 (4.7%) | 6/86 (7.0%) |
| Middle and high[b] | 82/108 (95.3%) | 80/86 (93.0%) |
| Smoking | | |
| Yes | 11/189 (5.8%) | 10/189 (5.3%) |
| Quit | 12/189 (6.3%) | 11/189 (5.8%) |
| No | 166/189 (87.8%) | 168/189 (88.9%) |
| Alcohol | 1/187 (0.5%) | 5/187 (2.7%) |
| Method of conception | | |
| Natural | 180 (92.8%) | 172/192 (89.6%) |
| IUI and/or ovulation induction | 8 (4.1%) | 10/192 (5.2%) |
| IVF/ICSI | 6 (3.1%) | 10/192 (5.2%) |
| Maternal medical history | | |
| Diabetes mellitus | 2 (1.0%) | 4 (2.1%) |
| Gestational diabetes | 11 (5.7%) | 10 (5.2%) |
| Renal disease | 4 (2.1%) | 2 (1.0%) |
| Inflammatory bowel disease | 2 (1.0%) | 0 (0%) |
| Thyroid disease | 11 (5.7%) | 7 (3.6%) |
| Chronic hypertension | 3 (1.5%) | 3 (1.6%) |
| Systemic lupus erythematosus | 2 (1.0%) | 0 (0%) |
| Cardiac disease | 3 (1.5%) | 7 (3.6%) |
| Obstetric history | | |
| *Parity* | | |
| 1 | 132 (68.0%) | 140 (72.5%) |
| 2 | 43 (22.2%) | 42 (21.8%) |
| ≥3 | 19 (9.8%) | 11 (5.7%) |
| *Number of previous spontaneous preterm births (22$^{+0}$ to 37$^{+0}$)* | | |
| 1 | 175 (90.2%) | 177 (91.7%) |
| ≥2 | 19 (9.8%) | 16 (8.3%) |
| *Number of midtrimester fetal losses (16$^{+0}$ to 21$^{+6}$)* | | |
| 1 | 13 (6.7%) | 2 (1.0%) |
| ≥2 | 1 (0.5%) | 0 (0%) |
| *Number of therapeutic abortions* | | |
| 1 | 9 (4.6%) | 10 (5.2%) |
| ≥2 | 4 (2.0%) | 7 (3.6%) |
| *Number of miscarriages and ectopic pregnancies (<16$^{+0}$)* | | |
| 1 | 45 (23.2%) | 43 (22.3%) |
| ≥2 | 28 (14.4%) | 28 (14.5%) |
| Risk factors for preterm birth | | |
| History of cervical surgery (conisation/LLETZ) | 13/191 (6.8%) | 7/190 (3.7%) |
| History of uterine surgery (e.g., myomectomy) | 6/191 (3.1%) | 2/191 (1.0%) |
| Cerclage in previous pregnancy | 13 (6.7%) | 12 (6.2%) |

(*Continued*)

**Table 1.** (Continued)

|  | Aspirin (*n* = 194) | Placebo (*n* = 193) |
|---|---|---|
| Uterus anomaly | 7/173 (4.0%) | 4/168 (2.4%) |
| Family history (mother/sister) of preterm birth | 8/95 (8.4%) | 6/105 (5.7%) |
| Short interpregnancy interval (<6 months from last pregnancy to conception) | 16 (8.2%) | 13 (6.7%) |
| Gestational age at randomisation–median (IQR) | $13^{+5}$ ($12^{+1}$ to $15^{+0}$) | $13^{+6}$ ($11^{+6}$ to $15^{+1}$) |
| Fetal sex (girl) | 98/192 (51.0%) | 89/193 (46.2%) |

[a]Primary school, prevocational secondary education (VMBO in Dutch).

[b]Senior general secondary education (HAVO in Dutch), preuniversity secondary education (VWO in Dutch), secondary vocational education (MBO in Dutch), higher professional education (HBO in Dutch), and university education (WO in Dutch).

ICSI, intracytoplasmic sperm injection; IUI, intrauterine insemination; IVF, in vitro fertilisation; LLETZ, large loop excision of the transformation zone.

Among all randomised women, serious adverse events occurred in 11 (5.4%) women allocated to aspirin and 11 (5.4%) women in the placebo group. There were no differences between treatment groups (S4 Table). None of these serious adverse events was considered to be associated with treatment allocation by the research team.

Prespecified subgroup analyses (S5 Table) demonstrated that there was a significant interaction between allocated treatment and the gestational age of the previous preterm birth ($p$ = 0.042). In women with a previous preterm birth <30 weeks of gestation, preterm birth occurred in 19.2% (15/78) of those allocated to aspirin and 32.6% (28/86) of those allocated to placebo (RR 0.59, 95% CI 0.23 to 1.02, $p$ = 0.059). There were no significant differences between treatment groups among women with a previous preterm birth at 30 weeks of gestation or later. No effect modification was found in subgroups based on the initiation of treatment, the use of progesterone, cervical length at asymptomatic screening, and the onset of the (earliest) previous spontaneous preterm birth.

## Discussion

In this multicentre, randomised, double-blinded, placebo-controlled trial including women with a previous spontaneous preterm birth <37 weeks of gestation of a singleton, the use of low-dose aspirin (80 mg) initiated from 8 to 16 weeks until 36 weeks of gestation did not significantly reduce the risk of a recurrent preterm birth of a singleton. The preterm birth rate was lower in the aspirin group compared to the placebo group, and, interestingly, the risk difference between treatment groups increased in women with good adherence to study medication. The treatment effect, however, still did not reach statistical significance. Given the lower than expected preterm birth rates, the present study was underpowered to assess the treatment effect related to preterm birth.

Preterm birth is a heterogeneous syndrome with multiple possible underlying pathophysiologic mechanisms [19]. Therefore, it is implausible that a single preventive measure will be able to prevent all preterm births. The administration of aspirin might be beneficial to a subset of women at risk. Subgroup analysis of the present study suggests that women with a previous spontaneous preterm birth <30 weeks of gestation might benefit more from low-dose aspirin prophylaxis than women with a spontaneous preterm birth later on in pregnancy, which could

**Table 2. Birth outcomes according to allocated treatment group.**

| | Aspirin (*n* = 194) | Placebo (*n* = 193) | RR[a]/Mean difference[b] (95% CI) | *p*-value |
|---|---|---|---|---|
| Preterm birth <37 weeks of gestation | 41 (21.2%) | 49 (25.4%) | 0.83 (0.58 to 1.20) | 0.323 |
| Spontaneous onset of preterm birth | 39 (20.1%) | 46 (23.8%) | 0.84 (0.58 to 1.23) | 0.376 |
| Indicated preterm birth | 2 (1.0%) | 3 (1.6%) | 0.66 (0.11 to 3.93) | 0.685[c] |
| Preterm birth <37 weeks of gestation in women with ≥80% adherence to therapy | 24/125 (19.2%) | 30/121 (24.8%) | 0.77 (0.48 to 1.25) | 0.291 |
| Preterm prelabour rupture of membranes (delivery <37 weeks of gestation) | 9 (4.6%) | 18 (9.3%) | 0.50 (0.23 to 1.08) | 0.077 |
| Gestational age at birth (weeks + days)–median (IQR) | $38^{+1}$ ($37^{+1}$ to $39^{+1}$) (95% CI $37^{+6}$ to $38^{+3}$) | $38^{+1}$ ($36^{+6}$ to $39^{+2}$) (95% CI $37^{+6}$ to $38^{+4}$) | - | 0.964 |
| Time between randomisation and birth (weeks + days)–mean (±SD) | $23^{+6}$ ($±4^{+1}$) | $24^{+1}$ ($±3^{+4}$) | $-0^{+1}$ ($-1^{+0}$ to $0^{+4}$) | 0.604 |
| Preterm birth ≤34 weeks | 18 (9.3%) | 17 (8.8%) | 1.05 (0.56 to 1.98) | 0.872 |
| Spontaneous onset of birth | 18 (9.3%) | 16 (8.3%) | 1.12 (0.59 to 2.13) | 0.732 |
| Indicated | 0 (0%) | 1 (0.5%) | - | - |
| Preterm birth ≤28 weeks | 7 (3.6%) | 5 (2.6%) | 1.39 (0.45 to 4.31) | 0.566 |
| Spontaneous onset of birth | 7 (3.6%) | 5 (2.6%) | 1.39 (0.45 to 4.31) | 0.566 |
| Indicated | 0 (0%) | 0 (0%) | - | - |
| Midtrimester fetal loss ($16^{+0}$ to $21^{+6}$ weeks) | 2 (1.0%) | 0 (0%) | - | - |
| Mode of birth | | | | |
| Spontaneous vaginal birth | 154 (79.4%) | 154 (79.8%) | 1.00 (0.90 to 1.10) | 0.920 |
| Assisted vaginal birth | 10 (5.2%) | 6 (3.1%) | 1.66 (0.62 to 4.47) | 0.318 |
| Cesarean delivery | 30 (15.5%) | 33 (17.1%) | 0.90 (0.58 to 1.42) | 0.663 |
| Postpartum hemorrhage | | | | |
| >500 mL | 50/192 (26.0%) | 51/188 (27.1%) | 0.96 (0.69 to 1.34) | 0.811 |
| >1,000 mL | 18/192 (9.4%) | 13/188 (6.9%) | 1.36 (0.68 to 2.69) | 0.383 |
| Birth weight (g)–mean (±SD) | 3,046 (±752) | 3,109 (±724) | −63.6 (−211.5 to 84.3) | 0.398 |
| Small for gestational age (<10th centile) | 16/193 (8.3%) | 15/193 (7.8%) | 1.07 (0.54 to 2.10) | 0.851 |

[a]RRs and the corresponding *p*-values were calculated using generalised linear regression analysis.

[b]Mean differences and the corresponding *p*-values were calculated using the independent samples *t* test.

[c]Due to the occurrence of ≤5 events, Fisher exact test was used to calculate the *p*-value.

CI, confidence interval; IQR, interquartile range; RR, relative risk; SD, standard deviation.

be a reflection of a different underlying cause of the preterm birth. Based on placental histological abnormalities, inflammatory causes of preterm birth are predominant among extreme preterm births before 28 weeks of gestation, gradually shifting to the predominance of placental insufficiency in later preterm births [20]. The possible stronger treatment effect of low-dose aspirin among women with a previous extreme preterm birth could indicate that low-dose aspirin may prevent certain subtypes of preterm birth through its anti-inflammatory properties.

There was a higher rate of adverse neonatal outcome in the aspirin group, mainly due to a higher mortality rate. The deaths included midtrimester fetal losses and extreme preterm births related to cervical insufficiency and intrauterine inflammation and 2 fetal losses diagnosed at 16 weeks of gestation. These deaths are inherent to the study population, and it seems unlikely that they are related to the treatment received. Previous reviews and meta-analyses did not find an increased perinatal mortality rate in women using aspirin for the prevention of preeclampsia [9,21].

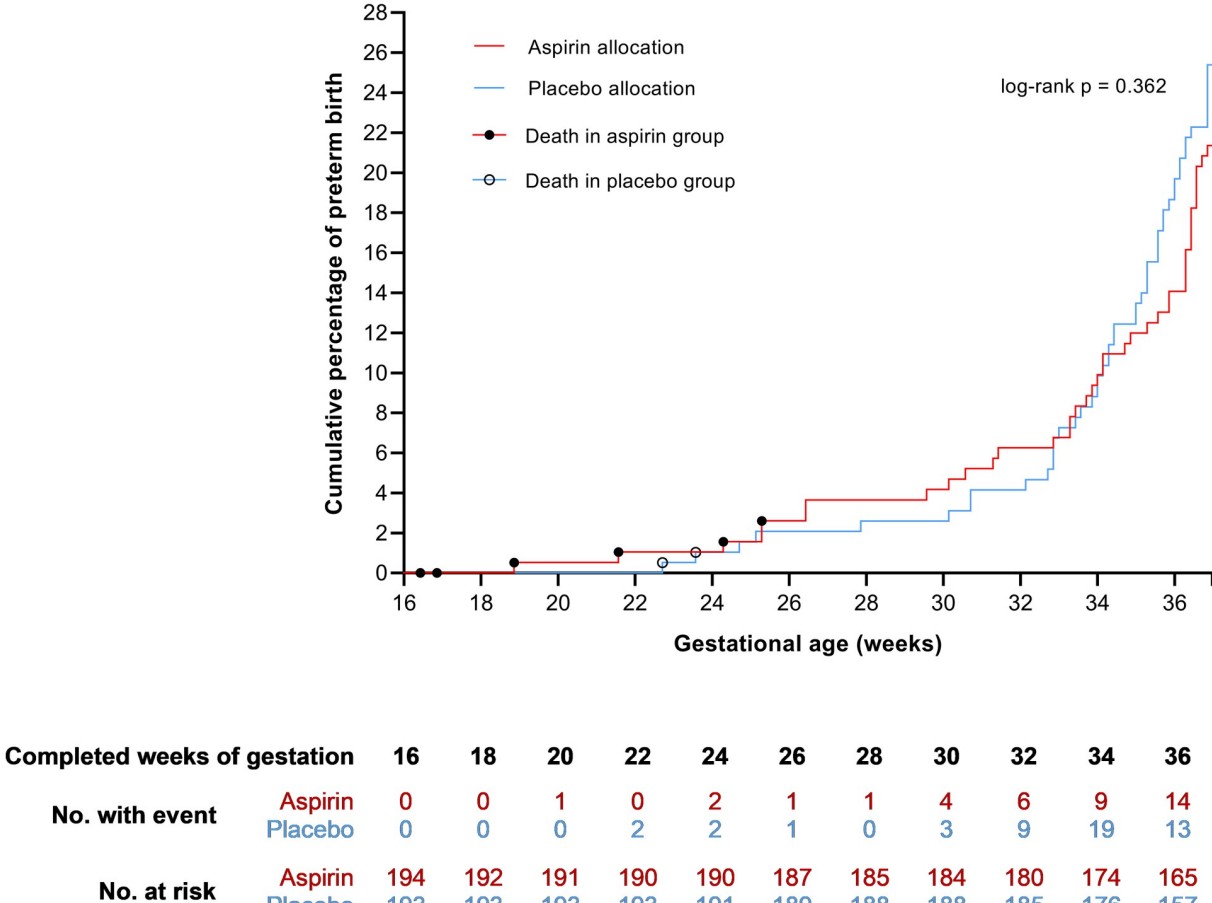

| Completed weeks of gestation | | 16 | 18 | 20 | 22 | 24 | 26 | 28 | 30 | 32 | 34 | 36 |
|---|---|---|---|---|---|---|---|---|---|---|---|---|
| **No. with event** | Aspirin | 0 | 0 | 1 | 0 | 2 | 1 | 1 | 4 | 6 | 9 | 14 |
| | Placebo | 0 | 0 | 0 | 2 | 2 | 1 | 0 | 3 | 9 | 19 | 13 |
| **No. at risk** | Aspirin | 194 | 192 | 191 | 190 | 190 | 187 | 185 | 184 | 180 | 174 | 165 |
| | Placebo | 193 | 193 | 193 | 193 | 191 | 189 | 188 | 188 | 185 | 176 | 157 |

**Fig 2. Kaplan–Meier curve for the cumulative incidence of preterm birth.**

Considering the dose-dependent effect of aspirin, the nonsignificant effect in our study could also be the result of a relatively low dose of aspirin. In the field of preeclampsia, there is some indirect evidence indicating aspirin ≥100 mg might be superior to doses <100 mg when started <16 weeks of gestation [22].

Secondary analyses of previous aspirin trials consistently reported lower rates of spontaneous preterm birth <37 weeks of gestation in the aspirin group compared to the control group with corresponding effect sizes ranging between 0.51 and 0.97 [11,12,23]. An individual participant data meta-analysis including 27,510 women at risk for preeclampsia also found a lower risk of spontaneous preterm birth <37 weeks of gestation (RR 0.93, 95% CI 0.86 to 0.996) in women using antiplatelet agents, mostly aspirin [13]. These results are in line with our findings. Therefore, we believe that our findings support the hypothesis that aspirin might be effective for the prevention of spontaneous preterm birth, even though our treatment effect did not reach statistical significance.

The ASPIRIN trial was the first randomised controlled trial evaluating the effect of aspirin with preterm birth as primary outcome [10]. They included 11,976 nulliparous women in low- and middle-income countries and found a reduction in preterm birth rate <37 and <34 weeks of gestation in the aspirin group. They did not distinguish between spontaneous and indicated preterm birth. As there were no differences in the principal causes of indicated preterm birth, they presumed that their results might reflect a reduction in spontaneous preterm birth.

**Table 3. Neonatal outcomes according to treatment group.**

| | Aspirin (*n* = 194) | Placebo (*n* = 193) | RR[b]/ Difference in medians[c] (95% CI) | *p*-value |
|---|---|---|---|---|
| Composite poor neonatal outcome | 9 (4.6%) | 5 (2.6%) | 1.79 (0.61 to 5.25) | 0.288 |
| Mortality | 6 (3.1%) | 2 (1.0%) | 2.99 (0.61 to 14.60) | 0.284[d] |
| BPD | 1 (0.5%) | 3 (1.6%) | 0.33 (0.04 to 3.16) | 0.372[d] |
| PVL > grade 1 | 0 (0%) | 0 (0%) | - | - |
| IVH > grade 2 | 1 (0.5%) | 0 (0%) | - | - |
| NEC > stage 1 | 1 (0.5%) | 0 (0%) | - | - |
| Retinopathy of prematurity | 1 (0.5%) | 2 (1.0%) | 0.50 (0.05 to 5.44) | 0.623[d] |
| Culture proven sepsis | 4 (2.1%) | 2 (1.0%) | 1.99 (0.37 to 10.74) | 0.424[d] |
| Mortality | | | | |
| Fetal death | 4 (2.1%) | 2 (1.0%) | 1.99 (0.37 to 10.74) | 0.685[d] |
| Neonatal death | 2 (1.0%) | 0 (0%) | - | - |
| Any hospital admission for neonatal indication | 80 (74.8%) | 84 (73%) | 1.02 (0.88 to 1.20) | 0.770 |
| Total days in hospital until 3 months corrected age–median (IQR)[a*] | 4 (2 to 13) (95% CI 3 to 6) | 5 (2 to 14) (95% CI 3 to 7) | −1 | 0.755 |
| NICU admissions | 13 (6.7%) | 11 (5.7%) | 1.18 (0.54 to 2.56) | 0.683 |
| Total days in the NICU until 3 months corrected age–median (IQR)[a*] | 12 (4 to 46) (95% CI 2 to 54) | 7 (2 to 58) (95% CI 2 to 82) | 5 | 0.560 |

[a]Of those neonates with an admission.

[b]RRs and the corresponding *p*-values were calculated using generalised linear regression analysis.

[c]The corresponding *p*-value was calculated using the Mann–Whitney U test.

[d]Due to the occurrence of ≤5 events, Fisher exact test was used to calculate the *p*-value.

*Of those neonates with an admission.

BPD, borderline personality disorder; CI, confidence interval; IQR, interquartile range; IVH, intraventricular hemorrhage; NEC, necrotising enterocolitis; NICU, neonatal intensive care unit; PVL, Panton–Valentine leukocidin; RR, relative risk; SD, standard deviation.

Furthermore, they found a significant reduction in stillbirth 16 to 20 weeks of gestation and perinatal mortality.

In contrast to other studies, the ASPRE trial did not find a decreased risk of spontaneous preterm birth in women taking aspirin to prevent preeclampsia [24]. The study population was selected based on a screening algorithm for preeclampsia. Even though there is some overlap in the underlying pathophysiologic mechanism between preterm birth and preeclampsia, the population at risk for preeclampsia probably differs from the population at risk for spontaneous preterm birth.

To the best of our knowledge, this is the first trial assessing the effect of aspirin for the prevention of preterm birth as the primary outcome in a high-risk population with a previous spontaneous preterm birth. The trial was well conducted and had no loss to follow-up and was of high quality owing to its double-blinded, placebo-controlled design and multicentre setting. The main limitation of our study was the sample size, which was underpowered. Our calculation was based on a preterm birth recurrence rate of 35%, which was, in fact, around 25%. The recurrence rate was based on Meis and colleagues assessing the effect of progesterone on preterm birth [8], and the potential RR reduction of 36% was estimated from the average treatment effect of aspirin in other pregnant populations. The second issue that should be addressed is the randomisation procedure. Despite proper randomisation, imbalances between treatment groups have occurred for some important baseline risk factors. No irregularities occurred in randomisation and masking, and, therefore, the randomisation imbalances are based on chance. The baseline imbalances were all in favour of the placebo group, indicating

**Table 4. Maternal outcomes according to treatment group.**

| | Aspirin (n = 194) | Placebo (n = 193) | RR[b]/ Difference in medians[c] (95% CI) | p-value |
|---|---|---|---|---|
| Maternal mortality | 0 (0%) | 0 (0%) | - | - |
| Maternal morbidity | | | | |
| Gestational diabetes | 15 (7.7%) | 15 (7.8%) | 1.00 (0.50 to 1.98) | 0.988 |
| Pregnancy-induced hypertension | 4 (2.1%) | 5 (2.6%) | 0.80 (0.22 to 2.92) | 0.751[d] |
| Preeclampsia/HELLP syndrome | 2 (1.0%) | 2 (1.0%) | 1.00 (0.14 to 6.99) | 1.000[d] |
| Eclampsia | 0 (0%) | 0 (0%) | - | - |
| Pulmonary edema | 0 (0%) | 0 (0%) | - | - |
| Thromboembolic disease | 0 (0%) | 0 (0%) | - | - |
| Placental abruption | 0 (0%) | 2 (1.0%) | - | - |
| Maternal self-reported symptoms (moderate to severe) | | | | |
| Vaginal bleeding | 5/106 (4.7%) | 7/116 (6.0%) | 0.78 (0.26 to 2.39) | 0.666 |
| Other bleeding[a] | 17/106 (16.0%) | 12/115 (10.4%) | 1.54 (0.77 to 3.01) | 0.222 |
| Gastrointestinal complaints | 14/105 (13.3%) | 19/115 (16.5%) | 0.81 (0.43 to 1.53) | 0.510 |
| Days of hospital admission during pregnancy for any reason–median (IQR) | 2 (1 to 3) (95% CI 1 to 2) | 2 (1 to 2) (95% CI 1 to 2) | - | 0.709 |
| Hospital admissions for vaginal bleeding during pregnancy | 10 (5.2%) | 9 (4.7%) | 1.11 (0.46 to 2.66) | 0.823 |
| Hospital admissions for threatened preterm labour during pregnancy | 33 (17.0%) | 41 (21.2%) | 0.80 (0.53 to 1.21) | 0.291 |
| Course of steroids for fetal lung maturity | 22/193 (11.4%) | 32/192 (16.7%) | 0.68 (0.41 to 1.13) | 0.140 |
| Tocolytic therapy | 19 (9.8%) | 28 (14.5%) | 0.67 (0.39 to 1.16) | 0.154 |
| Interventions during pregnancy | | | | |
| Progesterone | 136 (70.1%) | 129 (66.8%) | 1.05 (0.92 to 1.20) | 0.490 |
| Cerclage | 25 (12.9%) | 11 (5.7%) | 2.26 (1.15 to 4.47) | 0.019 |
| Pessary | 1 (0.5%) | 2 (1.0%) | 0.50 (0.05 to 5.44) | 0.623[d] |
| Bacterial vaginosis treated with antibiotics | 12 (6.2%) | 21 (10.9%) | 0.57 (0.29 to 1.12) | 0.104 |
| Urinary tract or genital infections treated with antibiotics | 6 (3.1%) | 15 (7.8%) | 0.40 (0.16 to 1.00) | 0.051 |

[a]Types of bleeding: anal bleeding, epistaxis, prolonged wound bleeding, or gingival bleeding.

[b]RRs and the corresponding p-values were calculated using generalised linear regression analysis.

[c]The corresponding p-value was calculated using the Mann–Whitney U test.

[d]Due to the occurrence of ≤5 events, Fisher exact test was used to calculate the p-value.

CI, confidence interval; HELLP, hemolysis, elevated liver enzymes, and low platelets; IQR, interquartile range; RR, relative risk; SD, standard deviation.

that the aspirin group had a higher baseline risk of preterm birth, which could have potentially biased results and diminished a possible treatment effect.

The present evidence does not support the implementation of low-dose aspirin for the prevention of spontaneous preterm birth; however, a modest effect of low-dose aspirin cannot be excluded. To evaluate the possible beneficial effect of low-dose aspirin for the prevention of spontaneous preterm birth, we encourage researchers to perform large placebo-controlled trials in different populations at risk, e.g., women with a previous spontaneous preterm birth (e.g., before 30 weeks of gestation) and nulliparous women. In light of current discussions on the optimal dose of aspirin, these trials should consider including 3 treatment arms: placebo, aspirin 75 to 80 mg, and aspirin 150 mg. As adherence to medication plays an important role in the clinical treatment effect of aspirin [25,26], detailed recording of medication adherence in such trials is required, and efforts should be made to enhance women's adherence. Furthermore, these trials should include short- as well as long-term outcomes to evaluate treatment effect and safety [27,28].

In conclusion, the present trial did not show a significant reduction of recurrent preterm birth of a singleton in women who used low-dose aspirin (80 mg) from 8 to 16 weeks until 36 weeks of gestation compared to placebo. A modest effect of aspirin, or an effect in a subset of women (e.g., with a previous early preterm birth), cannot be excluded with the current study.

## Supporting information

**S1 Fig. Maternal self-reported symptoms.**
(PDF)

**S1 Table. Inclusions per participating centre.**
(PDF)

**S2 Table. Medication use according to treatment group.**
(PDF)

**S3 Table. Additional neonatal outcomes according to treatment group.**
(PDF)

**S4 Table. List of serious adverse events.**
(PDF)

**S5 Table. Prespecified subgroup analyses.**
(PDF)

**S1 Appendix. Comparison to study protocol.**
(PDF)

**S2 Appendix. CONSORT checklist.**
(PDF)

**S3 Appendix. Definitions of outcomes.**
(PDF)

**S4 Appendix. Calculation of medication adherence.**
(PDF)

**S5 Appendix. Statistical analysis plan.**
(PDF)

## Acknowledgments

We would like to thank all participants, participating centres, research nurses, and the NVOG Consortium staff for their contribution to this trial.

Furthermore, we are grateful for the contributions of the members of the APRIL Study Group: Karlijn C Vollebregt (Department of Obstetrics and Gynaecology, Spaarne Hospital, Haarlem, the Netherlands), Elisabeth MA Boormans (Department of Obstetrics and Gynaecology, Meander Medical Centre, Amersfoort, the Netherlands), Henk A Bremer (Department of Obstetrics and Gynaecology, Reinier de Graaf Hospital, Delft, the Netherlands), Esther Tuinman (Department of Obstetrics and Gynaecology, Treant Care Group, Bethesda Hospital, Hoogeveen, the Netherlands), Josje Langenveld (Department of Obstetrics and Gynaecology, Zuyderland Medical Centre, Heerlen, the Netherlands), Flip van der Made (Department of Obstetrics and Gynaecology, Franciscus & Vlietland Hospital, Rotterdam, the Netherlands), Robbert JP Rijnders (Department of Obstetrics and Gynaecology, Jeroen Bosch Hospital, Den Bosch, the Netherlands), Huib AAM van Vliet (Department of Obstetrics and Gynaecology,

Catharina Hospital, Eindhoven, the Netherlands), Liv M Freeman (Department of Obstetrics and Gynaecology, Ikazia Hospital, Rotterdam, the Netherlands), Wilma M Monincx (Department of Obstetrics and Gynaecology, Sint Antonius Hospital, Utrecht, the Netherlands), Judith Blaauw (Department of Obstetrics and Gynaecology, Ommelander Hospital Group Groningen, Groningen, the Netherlands), Ineke Krabbendam (Department of Obstetrics and Gynaecology, Hospital Gelderse Vallei, Ede, the Netherlands), Rafli van de Laar (Department of Obstetrics and Gynaecology, VieCuri Medical Centre, Venlo, the Netherlands), Marieke FG Verberg (Department of Obstetrics and Gynaecology, Medical Spectrum Twente, Enschede, the Netherlands), and Hubertina CJ Scheepers (Department of Obstetrics and Gynaecology, Maastricht University Medical Centre, Maastricht, the Netherlands).

## Author Contributions

**Conceptualization:** Marjon A. de Boer, Laura Visser, Tobias A. J. Nijman, Marieke A. C. Hemels, Christiana N. Naaktgeboren, Marijke C. van der Weide, Ben W. Mol, Christianne J. M. de Groot, Martijn A. Oudijk.

**Data curation:** Anadeijda J. E. M. C. Landman, Laura Visser, Christiana N. Naaktgeboren, Marijke C. van der Weide, Judith O. E. H. van Laar, Dimitri N. M. Papatsonis, Mireille N. Bekker, Joris van Drongelen, Mariëlle G. van Pampus, Marieke Sueters, David P. van der Ham, J. Marko Sikkema, Joost J. Zwart, Anjoke J. M. Huisjes, Marloes E. van Huizen, Gunilla Kleiverda, Janine Boon, Maureen T. M. Franssen, Wietske Hermes, Harry Visser, Christianne J. M. de Groot, Martijn A. Oudijk.

**Formal analysis:** Anadeijda J. E. M. C. Landman, Christiana N. Naaktgeboren, Marijke C. van der Weide.

**Funding acquisition:** Marjon A. de Boer, Laura Visser, Tobias A. J. Nijman, Ben W. Mol, Christianne J. M. de Groot, Martijn A. Oudijk.

**Investigation:** Anadeijda J. E. M. C. Landman, Marjon A. de Boer, Laura Visser, Judith O. E. H. van Laar, Dimitri N. M. Papatsonis, Mireille N. Bekker, Joris van Drongelen, Mariëlle G. van Pampus, Marieke Sueters, David P. van der Ham, J. Marko Sikkema, Joost J. Zwart, Anjoke J. M. Huisjes, Marloes E. van Huizen, Gunilla Kleiverda, Janine Boon, Maureen T. M. Franssen, Wietske Hermes, Harry Visser, Christianne J. M. de Groot, Martijn A. Oudijk.

**Methodology:** Marjon A. de Boer, Marieke A. C. Hemels, Christiana N. Naaktgeboren, Marijke C. van der Weide, Christianne J. M. de Groot, Martijn A. Oudijk.

**Project administration:** Anadeijda J. E. M. C. Landman, Laura Visser.

**Supervision:** Marjon A. de Boer, Ben W. Mol, Christianne J. M. de Groot, Martijn A. Oudijk.

**Validation:** Anadeijda J. E. M. C. Landman, Marjon A. de Boer, Marijke C. van der Weide, Martijn A. Oudijk.

**Visualization:** Anadeijda J. E. M. C. Landman.

**Writing – original draft:** Anadeijda J. E. M. C. Landman, Marieke Sueters.

**Writing – review & editing:** Anadeijda J. E. M. C. Landman, Marjon A. de Boer, Laura Visser, Tobias A. J. Nijman, Marieke A. C. Hemels, Christiana N. Naaktgeboren, Marijke C. van der Weide, Ben W. Mol, Judith O. E. H. van Laar, Dimitri N. M. Papatsonis, Mireille N. Bekker, Joris van Drongelen, Mariëlle G. van Pampus, Marieke Sueters, David P. van der Ham, J. Marko Sikkema, Joost J. Zwart, Anjoke J. M. Huisjes, Marloes E. van Huizen,

Gunilla Kleiverda, Janine Boon, Maureen T. M. Franssen, Wietske Hermes, Harry Visser, Christianne J. M. de Groot, Martijn A. Oudijk.

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
