## [Editor Report · Decision Letter 0]

23 Jun 2021

Dear Dr Landman, 

Thank you for submitting your manuscript entitled "Aspirin for the Prevention of Recurrent Spontaneous Preterm Labor (APRIL): a randomized controlled trial" for consideration by PLOS Medicine.

Your manuscript has now been evaluated by the PLOS Medicine editorial staff and I am writing to let you know that we would like to send your submission out for external peer review.

Kind regards,

Louise Gaynor-Brook, MBBS PhD

Associate Editor

PLOS Medicine

---

## [Decision Letter · Decision Letter 1]

15 Sep 2021

Dear Dr. Landman,

Thank you very much for submitting your manuscript "Aspirin for the Prevention of Recurrent Spontaneous Preterm Labor (APRIL): a randomized controlled trial" (PMEDICINE-D-21-02759R1) for consideration at PLOS Medicine. 

Your paper was evaluated by four independent reviewers, including a statistical reviewer, and was discussed among all the editors here and with an academic editor with relevant expertise. The reviews are appended at the bottom of this email and any accompanying reviewer attachments can be seen via the link below:

[LINK]

In light of these reviews, I am afraid that we will not be able to accept the manuscript for publication in the journal in its current form, but we would like to consider a revised version that addresses the reviewers' and editors' comments. Obviously we cannot make any decision about publication until we have seen the revised manuscript and your response, and we plan to seek re-review by one or more of the reviewers. 

We expect to receive your revised manuscript by Oct 06 2021 11:59PM. Please email us (plosmedicine@plos.org) if you have any questions or concerns.

We look forward to receiving your revised manuscript. 

Sincerely,

Louise Gaynor-Brook, MBBS PhD

Associate Editor 

PLOS Medicine

plosmedicine.org

Comments from the Academic Editor:

Well conducted, well written, good supplementary material and limitations clearly presented. Two minor points:

Women were excluded if they had another indication for aspirin. What are the indications for aspirin in the Netherlands? Interestingly some trial participants had conditions that would indicate aspirin according to UK/US guidelines (eg chronic hypertension).

Dose could be discussed in the discussion - some studies have advocated 150mg rather than 75 or 80mg.

General comments:

Please replace the term “compliance” with “adherence” where it is used to refer to treatment adherence.

Throughout the paper, please adapt reference call-outs to the following style: "... rate of preterm birth [9,10]." (noting the absence of spaces within the square brackets).

Data availability:

PLOS Medicine requires that the de-identified data underlying the specific results in a published article be made available, without restrictions on access, in a public repository or as Supporting Information at the time of article publication, provided it is legal and ethical to do so. If the data are owned by a third party but freely available upon request, please note this and state the owner of the data set and contact information for data requests (web or email address). Please note that a study author cannot be the contact person for the data. If the data are not freely available, please describe briefly the ethical, legal, or contractual restriction that prevents you from sharing it, and include an appropriate contact (web or email address) for inquiries (again, this cannot be a study author).

Title: Please revise your title according to PLOS Medicine's style. Please place the study design in the subtitle (ie, after a colon). We suggest ‘Evaluation of Aspirin for the Prevention of Recurrent Spontaneous Preterm Labor (the APRIL study): a multicenter, randomized, double-blinded, placebo-controlled trial” or similar

Abstract:

Please report your abstract according to CONSORT for abstracts, following the PLOS Medicine abstract structure (Background, Methods and Findings, Conclusions) http://www.consort-statement.org/extensions?ContentWidgetId=562

Abstract Methods and Findings:

Please provide brief demographic details of the study population (e.g. sex, age, ethnicity, etc)

Line 96 - please define RRR at first use 

Line 97 - please replace ‘compliant’ with another term

Please state that analysis was intention to treat.

Please provide the actual number of events in addition to % and RRR

Please specify what is meant by ‘poor neonatal outcome’ 

Please include a summary of adverse events.

In the last sentence of the Abstract Methods and Findings section, please describe 2-3 of the main limitations of the study's methodology.

Abstract Conclusions:

Please begin your Abstract Conclusions with "In this study, we observed ..." or similar, to summarize the main findings from your study. Please emphasize what is new and address the implications of your study, being careful to avoid assertions of primacy. 

Author Summary:

In the final bullet point of ‘What Do These Findings Mean?’, please describe the main limitations of the study in non-technical language.

Introduction:

Line 127 - Please temper assertions of primacy by adding ‘to the best of our knowledge’ or similar. 

Methods:

Thank you for providing a CONSORT checklist. Please add the following statement, or similar, to the Methods: "This study is reported as per the Consolidated Standards of Reporting Trials (CONSORT) checklist (S1 Checklist)." When completing the checklist, please use section and paragraph numbers, rather than page numbers which will likely no longer correspond to the appropriate sections after copy-editing.

Thank you for providing your prospective protocol and statistical analysis plan. Please make sure that the Methods section transparently describes when analyses were planned, and if/when reported analyses differed from those that were planned. Changes in the analysis-- including those made in response to peer review comments-- should be identified as such in the Methods section of the paper, with rationale. If a reported analysis was performed based on an interesting but unanticipated pattern in the data, please be clear that the analysis was data-driven.

Results

Please incorporate Fig S1 into the main paper. 

Discussion:

Please present and organize the Discussion as follows: a short, clear summary of the article's findings; what the study adds to existing research and where and why the results may differ from previous research; strengths and limitations of the study; implications and next steps for research, clinical practice, and/or public policy; one-paragraph conclusion.

Please remove all subheadings within your Discussion e.g. Strengths and limitations

Line 322 - Please temper assertions of primacy by adding ‘to the best of our knowledge’ or similar. 

Line 340 - please define IPDMA at first use

Figures:

When a p value is given, please specify the statistical test used to determine it in the respective figure legend (including those in Supporting Information files). 

Tables:

When a p value is given, please specify the statistical test used to determine it in the respective table legend (including those in Supporting Information files). 

Please define all abbreviations used in the table legend of each table (including those in Supporting Information files).

References:

Please ensure that journal name abbreviations match those found in the National Center for Biotechnology Information (NCBI) databases, and are appropriately formatted and capitalised.

Please also see https://journals.plos.org/plosmedicine/s/submission-guidelines#loc-references for further details on reference formatting. 

Supplementary files: 

Please see https://journals.plos.org/plosmedicine/s/supporting-information for our supporting information guidelines. 

Comments from the reviewers:

Reviewer #1: Statistical review

This paper reports a RCT comparing low-dose aspirin to placebo for reducing pre-term birth. The methods and results are well reported. I have only some minor comments.

1. Abstract - I am unsure of whether featuring the subgroup analysis predominantly in the abstract is appropriate given that it is described as hypothesis generating and it is one of several subgroup analyses done.

2. Methods, outcomes - I would recommend that the paper lists all secondary outcomes in the main paper (although if complex the full definitions could be kept in the supplementary material).

3. Statistical analysis, Line 201 - I would recommend clarifying this is two-sided alpha.

4. Statistical analysis, I would recommend that the method used to get relative risk reduction is mentioned here. If any covariates were adjusted for in the analysis, I would mention them here.

5. Statistical analysis, for the Bayesian analysis there is not enough information given - presumably an informative prior was used corresponding to 36% and 23% in the two arms, but how informative (i.e. what Beta distribution was used). Although in the SAP I would summarise it in the main paper.

6. Statistical analysis, I would add what subgroups were pre-specified.

7. Line 256 - perhaps add 'indicating no strong evidence of difference between arms' after the Bayes factor result?

8. Line 273 - I would edit this to 'no significant differences'.

9. Methods/Results - I note the protocol mentions an economic analysis but this is not included here. It might be good to somewhere say that this will be reported elsewhere (or that it was planned for but not done).

James Wason

Reviewer #2: Nicely done paper. Unfortunately, as you point out, the study was underpowered to make a firm finding.

One question, if you did statistical re-analysis with the projected preterm delivery rate of 35% and risk reduction of 36% would your study show statistical/possible clinical significance?

Reviewer #3: This RCT of women with a prior SPTB comparing low dose aspirin (80mg) to placebo in women with a history of preterm birth. The project was well conducted with appropriate provision for masking and ascertaining outcomes. The paper is well written and the conclusions are well supported. Similarly they adhere to the CROWN guidance and provide rich supplementary materials. The following comments are designed to improve the manuscript.

Major:

1. The largest issue with this trial is the exceedingly ambitious power analysis. They assume a baseline rate from a study conducted with an American predominantly African-American population of whom almost have had 2 prior preterm deliveries. On top of this they assume an effect size of 35% when the prior literature had suggested a 10 to 20%. Finally they set the power at 80%. It is therefore not surprising that the out come is a trial in which the OR is directionally correct but not statistically significant. This is compounded by the fact as they describe there is a lack of parity in the two groups with the Aspirin group having higher risk.

2. Throughout the manuscript- p-Values are not kept to 2 significant digits. 

3. Preterm birth is the primary outcome yet the method of deriving this outcome is not clear established. Was a first trimester ultrasound required? How were discrepancies in LMP and due date by ultasound handled.

Minor:

1. Introduction: You do not need to state that "The remaining 65%" as this is inferred from the prior sentence. 

2. Intervention: UMC and AMC are unclear abbreviations. 

3. Methods: " The most important" Should be removed. This is a value judgement. 

4. Results: "The rate of women ....(n=14 , 7.2%) This number disagrees with the table that reports 13. Please decide which is correct. 

5. Table 1: please define education level and "quitted" should be quit or prior

Reviewer #4: I have read this paper in detail , now several times. It is an excellent randomised controlled trial where the authors have faced the recurrent challenges of a less than expected outcome rate and therefore as they say the study in the final analysis is underpowered. Also the study demonstrates that randomisation does not result in perfectly balanced groups. The baseline imbalances were all in favour of the placebo group and therefore this could mask the true effectiveness of the aspirin. I consider the study outstanding partly because of the way the authors have discussed and interpreted it. In particular they articulate very clearly the ambiguity of the results of their trial. This should serve as a model discussion piece. 

I would just ask for one minor revision; given the results, the underpowered nature of the study, the complexity of the arena and other challenges, a brief paragraph on the way forward with further investigations should be offered.

[LINK]

---

## [Decision Letter · Decision Letter 2]

18 Nov 2021

Dear Dr. Landman,

Thank you very much for re-submitting your manuscript "Evaluation of Low-dose Aspirin for the Prevention of Recurrent Spontaneous Preterm Labour (the APRIL study): a multicentre, randomised, double-blinded, placebo-controlled trial" (PMEDICINE-D-21-02759R2) for review by PLOS Medicine.

I have discussed the paper with my colleagues and the academic editor and it was also seen again by two reviewers. I am pleased to say that provided the remaining editorial and production issues are dealt with we are planning to accept the paper for publication in the journal.

[LINK]

We look forward to receiving the revised manuscript by Nov 25 2021 11:59PM.   

Sincerely,

Callam Davidson (on behalf of Louise Gaynor-Brook)

Associate Editor 

PLOS Medicine

plosmedicine.org

Requests from Editors:

In your Data Availability Statement, please provide some further detail as to the AMC policy on data sharing and how data will be made available to those who enquire. 

On line 136, the term "trend" is used to refer to a nonsignificant P value. The term trend should be used only when the test for trend has been conducted. Please revise accordingly.

Citations should be preceding punctuation, please update throughout.

Please remove the ‘Competing interests’, ‘Financial Disclosure Statement’, and ‘Data availability’ sections from the end of the main text; this information will be captured as metadata based on your responses to the submission form questions (so please ensure all relevant information is included in your answers).

Similarly, please remove the ‘Ethics approval’ section from the end of the main text as this information is already included in the methods section.

Table 3: Please confirm the meaning of the asterisk in table 3 or remove if it was included unintentionally.

Table S2 and Appendix S4: Please replace the term “compliance” with “adherence” where it is used to refer to treatment adherence.

Comments from Reviewers:

Reviewer #1: Thank you to the authors for addressing my previous comments well. I have no further issues to raise.

Reviewer #3: Thank you for addressing the comments so thoughtfully. I have no additional comments.

[LINK]

---

## [Editor Report · Decision Letter 3]

14 Dec 2021

Dear Dr Landman, 

On behalf of my colleagues and the Academic Editor, Dr. Sarah Stock, I am pleased to inform you that we have agreed to publish your manuscript "Evaluation of Low-dose Aspirin for the Prevention of Recurrent Spontaneous Preterm Labour (the APRIL study): a multicentre, randomised, double-blinded, placebo-controlled trial" (PMEDICINE-D-21-02759R3) in PLOS Medicine.

Before your manuscript can be formally accepted you will need to complete some formatting changes, which you will receive in a follow up email. Please be aware that it may take several days for you to receive this email; during this time no action is required by you. Once you have received these formatting requests, please note that your manuscript will not be scheduled for publication until you have made the required changes. Given our busy publication schedule, we are planning to publish your paper in early February 2022 (the exact date will be communicated to you once confirmed).

PRESS

Sincerely, 

Louise Gaynor-Brook, MBBS PhD 

Associate Editor, PLOS Medicine